# Interactive Learning: Unpacking the Influence of Computer Simulations on Students' Mathematical Modeling Processes

**Azin Sanjari** [1,2,*] **and Azita Manouchehri** [2]

1    Department of Mathematics and Statistics, Arkansas Tech University, Russellville, AR 72801, USA
2    Department of Teaching and Learning, The Ohio State University, Columbus, OH 43210, USA;
     manouchehri.1@osu.edu
*    Correspondence: ssanjaripirmahaleh@atu.edu

**Abstract:** Mathematical modeling and use of technology have been recognized as powerful vehicles for advancing mathematical thinking. The use of technology to aid students to gain understanding about real-world mathematical problems has long been a subject of inquiry. Using the extended Blum's modeling cycle as the theoretical framework, this study explores how computer simulations influence fifth-grade students' mathematical modeling processes. We investigated interpretations and perceptions of students regarding a situation model, and how these interpretations and perceptions changed as a result of exposure to the simulation. Observing students as they worked with three interactive computer simulations during task-based interviews, we found that animated graphics of the simulations helped the participants to visualize the problem and learn about the effect of different variables and their interactions on the outcomes of the simulations. Students explored simulation environments, ran simulations to observe outcomes of specific settings, and formed conjectures to describe the effect of simulation variables on the outcomes. Analyzing participants' activities and movements along the extended Blum modeling cycle, we propose a three-stage learning model composed of "exploration and discovery", "structured inquiry", and "synthesizing".

**Keywords:** mathematics education; mathematical modeling; interactive simulation; technology

## 1. Introduction

Within the past two decades, the use of mathematical modeling in curriculum and instruction has been endorsed by various scholars and professional organizations [1]. It has been suggested that mathematical modeling helps students to find mathematics relevant, useful, and applicable to real-world situations instead of seeing it from an abstract and isolated lens [2,3]. Additionally, modeling tasks tend to promote high cognitive demands that require students to explore, express, and generalize mathematical concepts [4]. Furthermore, modeling helps develop critical thinking skills [5] (p. 19). These activities help students to think creatively, ask questions that help them understand the model, and make an informed decision [6]. Finally, mathematical modeling promotes "classroom discourse" [5] (p. 19), which is crucial to children's social and mathematical development.

Although mathematical modeling as a content standard has been recognized as a central theme to secondary school mathematics for years, elementary and middle school students are typically not provided enough modeling opportunities that would help them develop skills in applying their knowledge to real-world problems [7–9]. It has been recognized that one approach to incorporating mathematical modeling in elementary grades could be the use of technology-based simulations that can assist students to visualize problems and explore their properties (e.g., Borromeo Ferri, 2018 [10]). Many stakeholders, such as the National Council of Teachers of Mathematics (NCTM) [11], have called attention to the importance of using technology in classrooms. The NCTM considers technology as an "essential tool for learning mathematics" in the 21st century, and proposes that students

must have access to technology in learning the content [11,12]. Researchers have considered how technology-enhanced learning affects outcomes amongst different student populations including gifted students and those with physical or intellectual disabilities. Much of this body of work suggests a positive relationship between students' academic performance and use of technology [12,13]. Williams [14] posited that technologies such as computers, audiovisual equipment, graphing calculators, and the like are educational resources that can drastically transform the learning process by creating an ideal environment for growth of understanding of content.

Despite this strong endorsement, there has been limited research on the impact of such tools and the type of support they might provide learners when examining mathematical modeling activities in elementary grades [15,16]. This study aimed to address this gap by considering how interactive computer simulations influence fifth-grade students' mathematical practices when considering modeling contexts. In particular, the research focused on ways that the learners interacted with the simulation and the impact of these interactions on their interpretations of tasks, their views of variables and parameters involved, and their predictions regarding the outcomes of changes to variable values. The goal was not to document individual learners' actions but to utilize evidence of their work towards building an empirically grounded model of how simulations were utilized by the participants in an effort to characterize phases of their modeling practices. One question guided data collection and analysis: How does the use of simulations influence participants' mathematical modeling processes?

## 2. Review of the Literature

Here we provide a summary of the literature on mathematical modeling and the role of technology in helping students develop mathematical models for real-world problems.

### 2.1. Mathematical Modeling

Models and modeling are recognized as an important part of scientific research in various fields including biology, engineering, and bioinformatics. English, Fox, and Watters [17] describe models as "conceptual systems used to construct, interpret, explain, and mathematically describe a situation" [5] (p. 17). Lesh and Fennewald [18] elaborated that a mathematical model is "a system for describing (or explaining or designing) another system for some specific purpose" (p. 7). Models break a real-world problem into pieces that are easy to study individually, express each piece in mathematical terms, and connect them in order to recreate the real-world problem as a mathematical system that mimics reality. Lesh and Doerr [19] suggest that models are conceptual systems that promote the development of conceptual understanding and mathematizing. Hence, models, which can have more than one representation, are descriptive solutions to a real-world problem. As a result, if the model cannot successfully predict the outcome of a real-world problem, it needs to be altered [20].

Blum and Borromeo Ferri [21] define mathematical modeling as "the process of translating between the real world and mathematics in both directions" (p. 45). They believe mathematical modeling aims to achieve the following: (a) help students to understand real-world phenomena; (b) support any form of mathematics learning, including but not limited to motivation, comprehension, and concept formation; (c) develop mathematical competency; and (d) contribute to a more accurate and descriptive view of mathematics. Therefore, mathematical modeling involves several mathematical practices such as critical thinking and the use of various concepts, all of which are cognitively demanding. The modeling process involves observing a real-world situation, conjecturing about the real-world situation, conducting mathematical analysis, obtaining results, and evaluating the model by comparing its result with the real-world situation [22].

The mathematical modeling cycle describes the modeling process [10,21,23] and provides a means for understanding how to trace individual thinking [10,24]. According to the modeling cycle [21,23], when encountering a real-world problem, the modeler initially

produces a situation model. The situation model is then simplified to a mathematical model by adding structure and considering conditions and variables and restricted parameters. This formal mathematical model is analyzed, outputting mathematical results, which are interpreted in terms of the real model. The results are validated as they are checked against the real-world conditions and constraints. This process iterates until a satisfactory model is obtained (Figure 1).

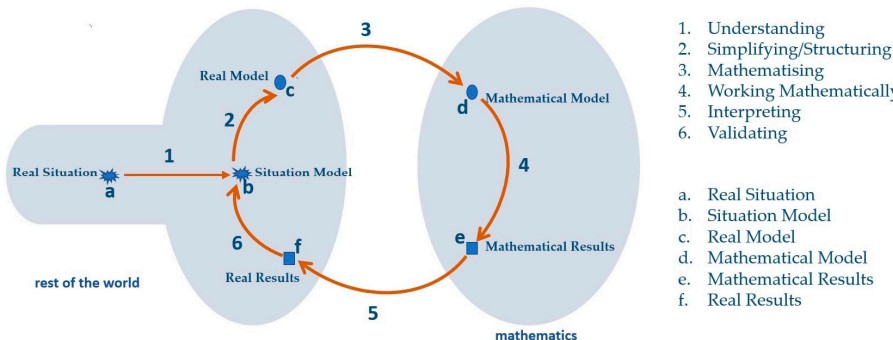

**Figure 1.** A schematic of modeling cycle according to Blum and Leiß, 2007 [23].

### 2.2. Use of Technology in Developing Mathematical Models of Complex Systems

Studies have shown that complex systems are good candidates for illustrating how modeling can be used to study a complicated problem with interacting components. According to Hmelo-Silver et al. [25], a complex system contains "multiple interrelated levels" with a dynamic structure. It is precisely the dynamic nature of interactions that makes complex systems difficult to understand. Lesh [26] proposed that the complex systems students typically encounter in their daily lives can be appropriately studied using relatively simple mathematical models. However, even these rather simple and daily-encountered problems become complex systems corresponding to complicated mathematical models if one aims to study them in detail. As opposed to deeper complex systems like those usually found in the biological sciences, English [27] defines "basic complex" systems for mathematical modeling problems designed for school mathematics. She identifies the five features of basic complex systems to include sets of interconnected parts, interactions among the parts, a system functioning as a whole, the relationship between the parts and the whole, and the effect of the parts of the system on each other.

### 2.3. Use of Computer Simulations for Mathematical Modeling

There is support that one way to engage learners in understanding complex systems is through the use of computer-based simulations [28]. According to Lee et al., computer simulations are "interactive software programs in which individuals explore new situations and complex relationships of dynamic variables that model real life" [29] (p. 902). Computer simulations enable students to form hypotheses on how input variables affect the simulation outputs, and test these hypotheses by changing the simulation parameters. Interactive computer simulations are computer modeling environments that contain a graphical interacting framework [28] and visualize the dynamic nature of real-world processes that affect the output. Research has shown that the dynamic visualization of computer simulations provides benefits over printed graphics [30].

Many researchers agree that using computer simulations improves student learning when real-world situations are not easily observable, or are simply impossible to generate and study in a traditional learning environment since they are not easily visualized [29,31]. In addition to providing visual representation of complex and complicated concepts, computer simulations allow students to explore complex and convoluted variable interactions [30]. Chilcott [32] reported that using simulations in the classroom helped students to be active and motivated in the learning process as running a simulation requires students' "problem solving" and "decision making" skills. Computer simulations help

students discover how different variables interact with each other and affect the output. Therefore, students engage in mathematical modeling by translating between the computer simulations and mathematics, instead of "translating between the real world and mathematics" [21] (p. 45).

## 3. Theoretical Framework

One of the most well-known modeling cycles is Blum and Leiß's six-stage schematic process [23] (Figure 1), which can be used to examine mathematical modeling from a cognitive perspective. According to this cycle, when encountering a real-world situation (a), the modeler tries to understand the model (1) and produces a situation model, which is a conceptual quantity in the mind of the modeler (b). Next, the modeler simplifies the model and adds structure to it by introducing conditions and variables (2), and a real model with internal and external components is obtained (c). Then, the modeler translates the developed model into mathematical language (3) and develops a mathematical model (d). The next step is analyzing the developed formal mathematical model (4). The output of this stage is production of mathematical results (e). These results can then be interpreted (5) in terms of the real model to obtain real results (f). These results are then validated (6) by checking them against the real-world results (b). Finally, the verified model can be shared with others, which is also known as communicating.

This model was extended by Greefrath [33] to account for the utility of technology in the modeling cycle (Figure 2). As illustrated in Figure 2, two stages are added to the original model to account for the use of technology, namely the computer model and computer results. In the extended model, a mathematical model results in a computer model, which produces computer results. The computer results are then translated to mathematical results. The mathematical results are then used as before to study the situation model, the real-world problem, or the mathematical model. Greefrath [33] emphasizes that digital tools, such as computers, allow us to solve real-world problems and mathematical models that could not be solved if digital tools did not exist.

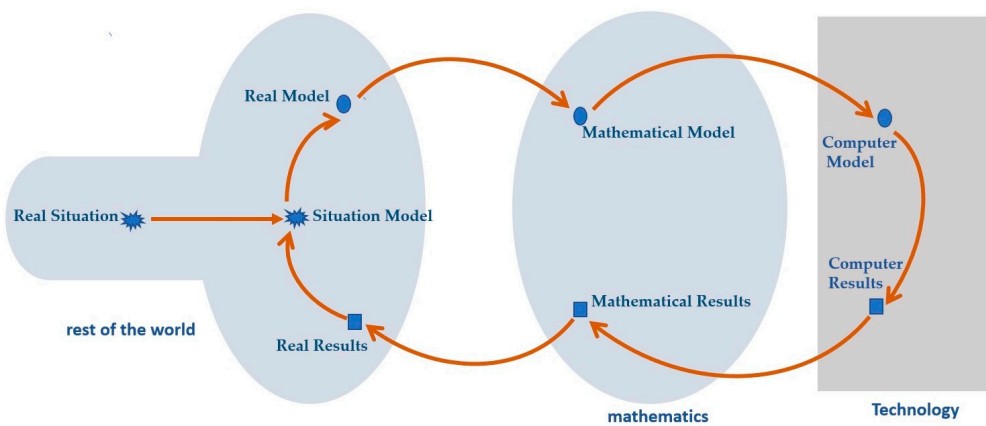

**Figure 2.** A schematic of the extended modeling cycle according to Greefrath, 2011 [33].

In the extended Blum modeling cycle, the modeler starts by considering a real-world problem and aims to develop or study a mathematical model describing it. Due to the large complex structures of mathematical models, various simulations are necessary to study a mathematical model. The goal is to design, develop, and implement a family of simulations such that, given the simulation results, students develop an understanding of the mathematics embedded in the problem and then reference and extend their understanding throughout the process. In doing so, they form meaningful hypotheses about the behavior of the complex system, describe the effect of parameters on the models, justify how their hypotheses are drawn from the simulation results, and contemplate how they can be further investigated.

The theoretical framework used in this study combines the extended mathematical modeling cycle and modeling activity diagrams. This hybrid strategy is motivated by Czocher [34]; however, the theoretical framework used in this study differs from the one used by Czocher [34] in order to account for the use of simulations, and is tailored for elementary school students. In particular, we define a set of activities that fully capture the modeling process of elementary school students and describe how their use of simulations to learn about real-world problems depends on the evolution of the "level of familiarity" of the student with the problem as they work with the simulation environment. The theoretical framework developed in this study explicitly acknowledges that the different uses of the simulation environment that are consecutively visited in a short period of time depend on the student's understanding of the problem and the goal they aim to achieve when running a simulation. More precisely, we differentiate if a simulation with a specific setting is run to visualize a general pattern of the system, learn about the interactions of several variables, or validate the student's prediction of the system behavior. These uses, their characteristics, and the frequency of their use based on the level of familiarity of the student with the problem can be easily studied within the proposed theoretical framework.

## 4. Methodology

### 4.1. Research Design and Data Collection

As is the tradition in research studies that target deep analysis of learning with the goal of theory building, the sample consisted of three fifth-grade students (10 to 11 years old) enrolled in an elementary school in the Midwest United States who volunteered to participate in the study. Selected participants had no prior experience working with computer simulations. All interviews were carried out as an after-school activity. The students were enrolled in different school districts, and they had different mathematical backgrounds.

A task-based semi-structured clinical interviewing approach was used for data collection. As defined by Maher and Sigley [35], task-based interviews are "interviews in which a subject or group of subjects talk while working on a mathematical task or set of tasks" (p. 579). One interview was conducted with each participant for each simulation. Each interview lasted approximately 35–50 min and students' interactions with an interactive simulation environment were documented.

All interviews were video-recorded and all artifacts produced during each interview were collected. During the interviews, the video camera focused on participants' written work and the computer screens they used when working with the simulation environment. To reveal the participants' thinking processes during the interviews, a think-aloud procedure was used. During the interviews, participants were routinely asked to clarify their answers or explain their reasoning and ideas to better understand their thinking without influencing what they did or leading them in a particular path. The interviewer's comments were restricted to asking clarifying questions, seeking explanations that could help interpret the learners' actions, and providing technical assistance when needed. These questions were not intended to alter the learners' actions or shape their interactions with the simulation environments.

### 4.2. Task Selection

The simulation environments and the tools they provide learners play an important role in determining their suitability for elementary students. In this study, we selected three interactive computer simulations that allowed students to observe an experiment under various simulation settings. Thereby, students were able to form an initial understanding about the problem, form a hypothesis about the behavior of the complex system, describe the effect of parameters on the models, justify how they explain the simulation results, and predict how they can be further investigated. The following criteria were considered for selecting simulations: (1) The number of variables should not be too large or too small so that students are neither lost or confused, nor dealing with a trivial problem. (2) The relationship between variables should not be significantly complicated or discontinuous.

(3) The simulation should mimic a tangible real-world problem so that students can make a connection between the problem and the real world.

The first simulation animates a cat chasing a mouse, where the mouse is running towards its home. The students are able to select the initial positions and the running speeds of the cat and mouse. They observe the movements of each animal, and whether the cat catches the mouse before the mouse reaches home (Figure 3). The goal of this problem is for students to be able to describe how initial positions and running speeds affect the final result (the cat catching the mouse, or the mouse successfully escaping the cat).

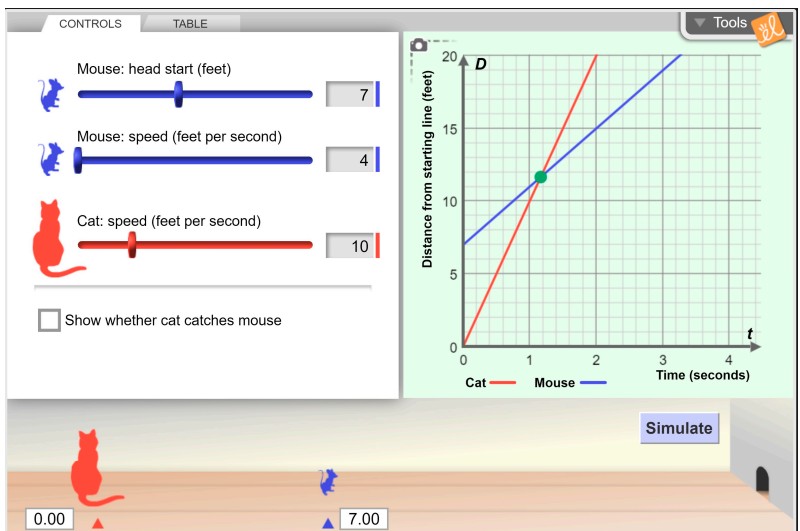

**Figure 3.** Interface of Cat and Mouse simulation "https://www.explorelearning.com/index.cfm?method=cResource.dspDetail&ResourceID=108 (accessed on 11 April 2019)".

The second simulation depicts a cannon shooting an object, e.g., a ball, in the air to land on a target on the ground. The students can adjust the height of the cannon, its angle of release, the initial speed of the ball being shot, and the location of the target. The goal is for students to be able to describe how these variables affect the location the ball hits the ground, and how a variable should be altered so that an overshot or undershot ball hits the target (Figure 4).

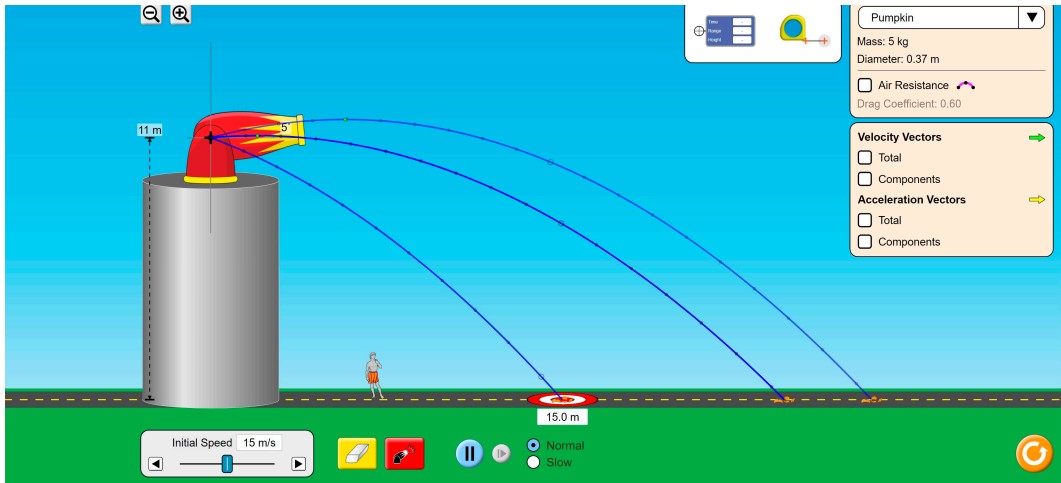

**Figure 4.** Interface of the Cannon Ball simulation "https://phet.colorado.edu/sims/html/projectile-motion/latest/projectile-motion_en.html (accessed on 11 April 2019)".

The third simulation is a set of darts that are thrown to a dartboard. The students can control the size of the board and the target on the board. The goal is for students to describe how each variable affects the ratio of darts hitting the target (Figure 5).

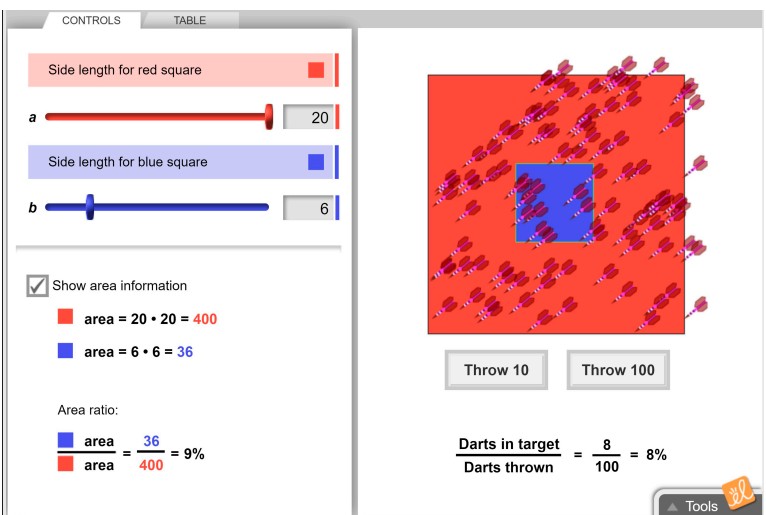

**Figure 5.** Interface of Dart Throwing simulation "https://www.explorelearning.com/index.cfm?method=cResource.dspView&ResourceID=138 (accessed on 11 April 2019)".

### 4.3. Data Analysis Process

Data analysis followed a three-stage process. First, videos of each of the interview sessions were transcribed and reviewed to identify prevalent themes. Transcripts were analyzed to identify "indicators" of modeling activities. Indicators highlighted student activities pertaining to working with the simulation or understanding the real-world problem modeled by the simulations. We coded activities and transitions relying on the modeling cycle [23] to trace participants' modeling processes. During the second stage, indicators were associated with the modeling activities and transitions along the modeling cycle [23] and extended modeling cycle [33]. For example, when students generated a list of simulation settings to test a conjecture, we identified this action as an indicator of the simplifying/structuring activity. Additional examples of indicators developed and used are provided in Tables 1 and S1. We observed that modifications to the extended modeling cycle are necessary to adequately track students' modeling processes. Therefore, we generalized and modified the extended modeling cycle based on students' interactions with the simulations. Activities of each student across each interview session were reviewed independently to make sure their modeling activities are properly grasped by our modified extended modeling cycle for all students across all simulations. All information was formatted as a spreadsheet to create an event map, containing the time interval, the indicator used, i.e., the participants' responses, the transcript or interaction with the simulation, and the modeling activity or transition of the student. In this study, modeling activities and transitions are the codes used for data analysis. Coded transcripts were reviewed by both authors to ensure validity (see Table S2 for examples). In several cases, students' body language, such as signs of understanding, was used to verify transitions between phases. Note that in this study an event denotes one student at an interview session working on a specific task. Students' interactions with the simulation environment were additionally used to associate their responses or interactions with the modeling cycle "indicators". Table 1 provides an example of an event map for one of the interview sessions.

**Table 1.** Example of an event map of an interview session, Cat and Mouse simulation.

| Time (Seconds) | | Modeling Activities (Code) | Evidence from Participants' Interactions with Simulation (Indicator) |
|---|---|---|---|
| 0:00:00 | 0:00:20 | Understanding | Student reads the problem |
| 0:00:20 | 0:00:40 | Understanding, Running | She tries to see how the simulation works, changing the angle (graphically), runs |
| 0:00:40 | 0:01:00 | Understanding, Running | Plays with height and angle (graphically) |
| 0:01:00 | 0:01:20 | Understanding, Running | Plays with height and angle (graphically) |
| 0:01:20 | 0:01:40 | Understanding, Running, Simplifying | Angle on 45 (middle), height on maximum |
| 0:01:40 | 0:02:00 | Running, Conditional | Runs and it does not hit, fixes all variables and reduces the speed |
| 0:02:00 | 0:02:20 | Understanding, Running, Simplifying | Fixes one variable and changes two others; runs each time to see the results |
| 0:02:20 | 0:02:40 | Running | Runs the new setting |
| 0:02:40 | 0:03:00 | Running, Simplifying | Fixes one variable and changes two others; runs each time to see the results |
| 0:03:00 | 0:03:20 | Understanding, Simplifying | Tries to understand the effect of different objects, fixing one variable at time |
| 0:03:20 | 0:03:40 | Understanding, Running, Simplifying | Maximizing and minimizing variable values, running each setting |
| 0:03:40 | 0:04:00 | Understanding, Running, Simplifying | Maximizing and minimizing variable values, running each setting |
| 0:04:00 | 0:04:20 | Understanding, Running, Simplifying | Maximizing and minimizing variable values, running each setting |
| 0:04:20 | 0:04:40 | Understanding, Running, Simplifying | Maximizing and minimizing variable values, running each setting |

In the third phase, modeling activity diagrams (MADs) [36] were generated to trace the participants' interactions with the simulation. These results were then used to capture students' activities based on the extended modeling cycle [33]. These key features and observations were then used to answer the research question. To generate MADs, modeling activities and phases of an extended modeling cycle were justified and re-defined to map participants' modeling activities to the cycle when they were working with interactive simulations. These new activities then served as a baseline to generate MADs and develop the new modeling cycle with added interactive simulation to all activities.

**5. Results**

In this section, we first describe how the simulation environment components and student activities were associated with states and transitions of the extended modeling cycle. Then, we share, by offering an example, how MAD analysis was used to trace the activity routes of participants when working with the simulations. Lastly, we illustrate how MAD analysis was utilized to investigate different stages of learning undertaken by the participants.

*5.1. Adapting the Extended Blum Modeling Cycle to Student Activities*

The phases and transitions the participants encountered during the interviews were described using a modeling cycle [23] with an added computer model [33]. These transitions and stages were then used to modify and justify the model for the purpose of the current study. Since the technology used in this study is interactive computer simulations, we adjusted the definitions of modeling cycle phases and transitions between phases according to the extended modeling cycle [33]. We observed indicators, such as designing a simulation,

that did not correspond to an activity or transition of the extended modeling cycle [33]. In this study, students did not develop computer programs to test their mathematical models. Instead, simulations replaced situation models by mimicking a real-world problem. We observed that running a simulation is an activity and is followed by evaluation of the simulation results. Furthermore, when students observed results inconsistent with their expectations, they reconciled their mathematical model with the new observations. Therefore, we modified the extended modeling cycle based on observed student activities during the interview sessions (Figure 6). These are described below.

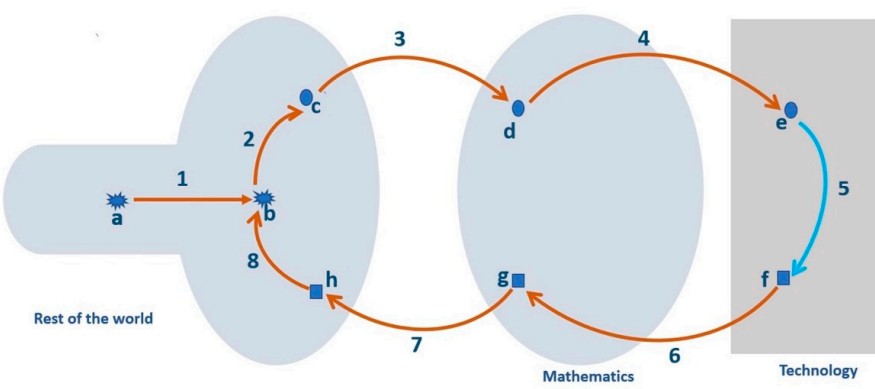

1. Understanding
2. Simplifying/Structuring
3. Mathematising
4. Designing Simulation by Programmer (Students do not visit this activity when they are using designed simulation)
5. Running
6. Experience Reconciliation
7. Interpreting
8. Validating

a. Real Situation
b. Simulation
c. Real Model
d. Mathematical Model
e. Computer Model
f. Computer Results
g. Evaluated Results
h. Real Results

**Figure 6.** The edited extended modeling cycle used in this study based on the extended modeling cycle [33]. In this study, the simulation replaces the situation model of [33], evaluated results replace the mathematical results of [33], and the transitions between modeling cycle activities of [23] are generalized for computer simulations by adding designing a simulation, running, and experience reconciliation.

**Understanding:** Given a real situation (i.e., the real-world problem), participants tried to map the simulation environment and its parameters with the real-world problem. We hereafter use the term "understanding" to denote the effort to explain and understand the simulation parameters based on real-life consideration.

**Simplifying/structuring:** Given the number of interacting variables in the simulation environment, participants often tried to impose a structure on the simulation settings they used (e.g., fixing the value of a variable or looking at the effect of a variable for its extreme values). We denote the process of a simulation design that students engage in to better learn about the simulation environment as "simplifying/structuring".

**Mathematizing:** Similar to the previous literature [23], we denote the students' efforts towards describing their observations or thinking in mathematical language as "mathematizing".

**Running:** An important activity in participants' interactions with the simulation environment was running simulations, which was present throughout the interview sessions. While it is not given an explicit name in the literature, due to its frequent presence we hereafter denote this process as "running".

**Experience reconciliation:** Given a set of computer results, participants often reconciled them with previous experiences and coordinated the effects of different variables based on the observations they previously made in order to arrive at an interpretation that accounted for all the computer results they had seen, in particular those most recently obtained. We hereafter denote this transition from computer model to evaluated results as "experience reconciliation". Note that we use the term "evaluated results" instead of "mathematical results". This stage is further described below.

**Interpreting:** The participants aimed to interpret, explain, understand, and justify the output of a simulation in terms of the real-world problem and previous simulation results. We hereafter denote the explanation and the effort of connecting evaluated results as "interpreting".

**Validating:** Given a question or prediction about the behavior of the simulation, participants ran a simulation to verify if their anticipated result might be true. We hereafter denote the process of interacting with the computer simulation to validate a predicted behavior or verify an answer as "validating".

In the following, we describe the extended modeling cycle phases that participants visited when they used interactive computer simulations to solve problems. Figure 6 depicts the extended modeling cycle according to defined phases and modeling activities in this study.

**Real situation**: In this study, the term "real situation" is used to describe the real-world problem related to the simulation. The students aimed to use the interactive simulation they were given in the context of a real-world problem.

**Simulation**: Instead of a situation model, being an interpretation of the real situation, participants had an interactive simulation to work with, which they connected to the real model. Therefore, participants used the simulation as a representation of the situation model. They used it to implement their designed settings to obtain a computer model that yielded computer results and used the simulation's graphical representation to connect the results to the real situation.

**Real model**: In the original definition of the modeling cycle [23], the real model is a model with internal and external variables. Here, given that the situation model was replaced with the simulation, we directly connect this phase to the simulation. We denote the real model as the case where the simulation model is considered under specific conditions, so that certain variables are more emphasized and that their effect can be studied. More precisely, a real model is the simulation under a purposeful screening lens, so that it can be studied in a specific aspect.

**Mathematical model**: As participants in this study were in the fifth grade and the simulation environments were structured, mathematizing did not involve solving complex equations or production of precise mathematical models that described how variables interact using functions. Rather, a mathematical model was a set of guidelines that justified certain simulation behaviors and provided strategies for how to alter the simulation settings to adjust the output in a desired way. These guidelines could be in the form of inferred rules between the variable values and the output; a collection of settings that depict a pattern; a statement about if and how two variables interact; or "if then inequality/pattern" forms that describe the effect of a setting change on the simulation behavior. For example, an "if then inequality/pattern" in the Cannon Ball simulation might state that "if we increase the initial height, then the range increases" or "if the height is zero, increasing the angle up to 45 degrees increases the range and increasing it more reduces the range". Similarly, in the Dart Throwing simulation, an "if then inequality/pattern" might state that "if we increase the area of both the target and dartboard so that the ratio remains untouched, then the probability of hitting the target is unaffected". Finally, in the Cat and Mouse simulation, an "if then inequality/pattern" might state that "if in a setting the cat does not catch the mouse, reducing the speed of the cat, increasing the head start, and increasing the speed of the mouse do not change the outcome".

**Computer model**: We use the term "computer model" to refer to the simulation environment under a specific input variable value so that it can be run. While the general terminology allows for actions such as computer programming, here, with the help of a graphical simulation environment, students can easily use the environment to specify a desired simulation setting that they want to evaluate.

**Computer results**: Computer results were the outputs of the simulations that were provided to the user. These outputs are the animated videos of the experiment, images, and graphs generated by the graphical interface of the simulation environment.

**Evaluated results**: Computer results were transformed into "evaluated results" and meaningful output variable values or relations that could be further used for learning about the simulation. For example, stating "the cat catches the mouse", "the location where the cat catches the mouse is X", "the range of the cannon ball is farther than the target distance", and "the number of darts hitting the target are more/less than the expected value" were all forms of evaluated results that participants used to describe computer results. Instead of writing an equation, reporting the value of a parameter, or calculating a quantity by hand based on the computer results, participants could immediately use the computer results to report the behavior of the simulation, and typically it was in the form of a statement or inequality. Therefore, in this study, we use the term "evaluated results" instead of "mathematical results".

**Real results**: Evaluated results form a basis to interpret how a simulation should behave and how the behavior connects to the actual real-world quantities. The real results could be used to form a statement about the simulation, adjust/validate the mathematical model, state one's belief about the constraints a valid mathematical model should satisfy, or predict how the simulation would behave given the current results and small changes to the settings. The real results were the information used to validate an understanding about the simulation.

### 5.2. Generating Modeling Activity Diagram (MAD)

MADs are graphical representations of an event (i.e., one student working on one task) that reveal participants' modeling activities in each time frame. A total of nine MADs were constructed to capture activities of all three participants during the interview sessions where they interacted with a simulation environment. Modeling activities versus time were plotted to trace participants' movements through identified transitions. Figure 7 provides an example of a MAD capturing one student's interactions with the Cannon Ball simulation. In this graph, the *y*-axis is the set of justified modeling activities (e.g., understanding, running, simplifying/structuring, mathematizing, experience reconciliation, interpreting, validating) and the *x*-axis denotes time measured in seconds.

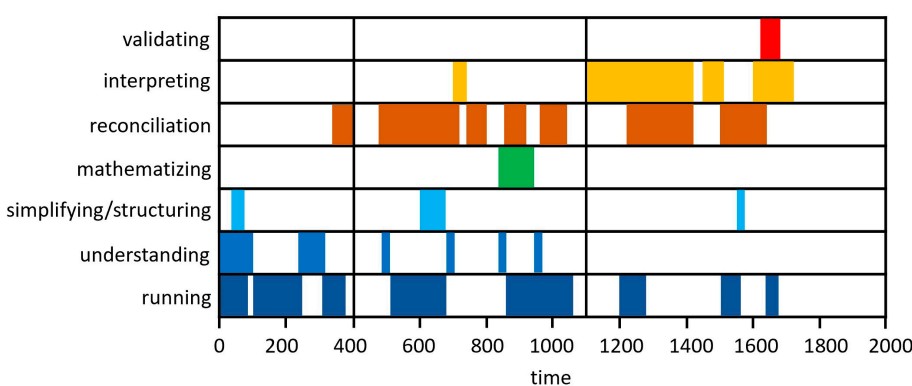

**Figure 7.** Example of a MAD created for one student's modeling activities.

We used 20 s time intervals to identify students' modeling activities. In Figure 7, the x- and y-axes denote time and modeling activities, respectively, and stars denote the modeling activity performed in the 20 s time interval. For example, "running the simulation" occurred continually throughout the entire session while mathematizing was visited only between the time interval of 780 s and 860 s of the interview session. Furthermore, validating was only visited in two consecutive time intervals of this interview, both towards the end of the interview session.

Analysis of MADs for each simulation evidenced that "running" the simulation was prominent and was performed to explore the simulation environment, test hypotheses, and answer the interviewer's questions. Continued reliance on the simulation environments and running models has been reported in previous studies [37]. However, there were

several modeling activities that were more clustered during specific times of the interview session. Validating was mostly observed towards the end of the simulation, while understanding happened towards the beginning. At the beginning, participants seemed to go back and forth between the computer model and computer results to run a new setting each time. They remained at the understanding and simplifying/structuring phases of the modeling process. This was most prominent at the beginning of their play time with the simulation environment and seemed to take up about 400–500 s across the different participants and simulations. During this stage participants explored the simulation to learn about the environment, considered the variables they could change, ways that the simulation results were presented, and what they might expect to see from the next trials. Following this phase, participants tended to move between the "experience reconciliation" activity and to a lesser extent the "mathematizing" and "interpreting the results" stages. During this phase, instead of simply playing with the simulation environment, participants deliberately chose a specific setting to serve a specific purpose. The significant number of occasions during which the participants constructed/expressed "if . . . then" statements supports the shift from random to deliberate use of the simulation. The settings the participants designed for their next simulation depended on their current observations, as well as their overall "batch design", i.e., a collection of simulations designed a priori before observing any of the results. The use of this strategy was mostly observed at the beginning of the interviews and the iterative approach was mostly utilized towards the end of the interviews.

Finally, in the last stage, which was mostly prevalent at the end of the interviews, participants relied on previous observations to make predictions about the behavior of the simulation under a specific setting or according to a particular change to the simulation. They intentionally adjusted settings to observe the occurrence of certain patterns, which they anticipated given their previous experiences with the simulation. Therefore, in the last stage participants visited the interpreting and validating stages of the modeling cycle.

### 5.3. A Three-Stage Model of Learning

Analyzing the MADs revealed three distinct phases in the participants' mathematical practices. Using the extended modeling cycle [33], the transcribed data, and the event map frequencies at different stages, we characterized these three phases as "exploration and discovery", "structured inquiry", and "synthesizing" (Figure 8). These three phases were compared against narratives of the sessions to assure they were descriptive of the data, participant activities, patterns among their responses, and ways that their modeling practices shifted as they worked with the simulation environments. Event narratives and participants' verbal expressions that disclosed their knowledge and thinking were used as main sources for explaining how the phases were different. Such moments included occasions when participants mentioned they did not know what to do or did not recognize what was happening, as well as times they indicated they did not know how to represent the evolution of the system or the ways that changes in variables affected the simulation.

**Exploration and discovery:** The first phase is called exploration and discovery. In this phase, participants focus on exploring the simulation environment, learn how to interact with it and use it, translate their formulation of the problem to match the simulation environment, and use the simulation to explore and discover different patterns and behaviors that result from these actions. This phase mostly included interacting with the computer model and computer results, and visualizing and learning more about the problem, as well as the simulation environment. For instance, at the beginning of their interaction with the simulation the participants tended to explore the options made available to them, the variables they could adjust, the outputs of various trials, and ways that they could use this body of information to guide their subsequent trials. Also, at this stage the participants tended to explore extreme cases with the goal to construct emerging patterns.

**Structured inquiry**: The second phase is called structured inquiry. While exploring the simulation during the open play time, participants noticed certain patterns and re-

lationships that motivated them to scrutinize what they had observed and to test them. Participants ran several settings to query the simulation for patterns and used the results to generate case-based scenarios. Such structured inquiries occurred either at the end of the play time or were provoked by the interviewer's questions.

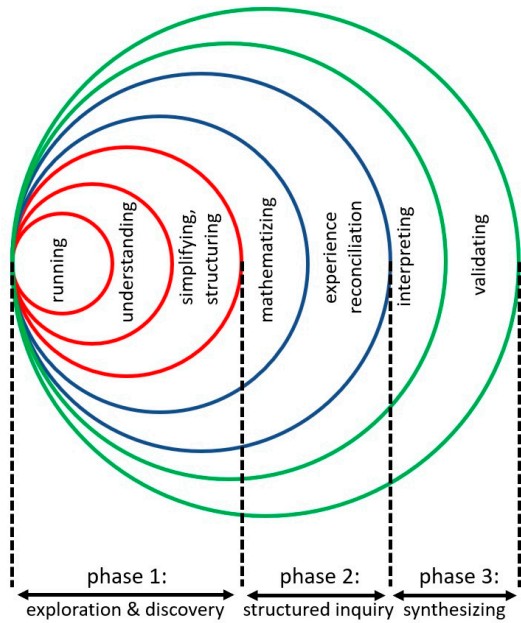

**Figure 8.** Student learning model with three stages.

In this phase, the participants also tended to systematically use the simulation environment as a means to collect data and make an inference. They used their collective results to further investigate, discuss, and discover relations between variables that could be used to establish general laws that govern the behavior of the system. Some of the indicators of inquiries of the participants were their reports of interesting patterns they did not know about or had just hypothesized and verified. Vignette 1 presents an illustrative example of how student A used the Cat and Mouse simulation environment (Figure 3) to observe a relationship between two variables: the head start of the mouse and the time it takes the cat to catch the mouse. She doubled the head start value and reported one of her observations: if the initial distance (head start) is doubled, the time it takes the cat to catch the mouse would double too.

Vignette 1

**Student A:** . . . also I realized that it would double such as . . . the first square, the second square . . . like it would double the squares [showing the square units in the graph] and then I tripled it and I got . . . umm . . . 12, 4 and 12 [12 for head start, 4 for mouse speed and 12 for cat speed] and I simulated and it did triple there too [showing the point that the cat catches the mouse] and I realize that it doubled again in how many squares . . . because it was two squares high and now it's four squares high and also it is 18 units [showing the y component of the intersection point] . . . So it doubled again

**Synthesizing:** The third phase is called synthesizing. Having a set of structured inquiry results, participants moved towards constructing simulation settings that were intentionally designed to verify a relation, answer the interviewer's questions, or validate their answers. These steps are labeled as mathematizing, which also contains synthesizing simulation settings for a specific purpose. In this phase, participants made progress in deciphering and identifying complex interactions amongst the variables. They could make predictions about the behavior of the simulation setting prior to running it and could explain why they were making certain predictions. In their explanations, they were

referring to previous settings they had tested or rules they had inferred in phase II. Vignette 2 presents an illustrative example of how student B inferred the outcome of the Cannon Ball simulation (Figure 4) when asked to predict, without running the simulation, how the results may change according to changes to the setting:

Vignette 2

**Interviewer:** So, now, I have a question. Your numbers are seven, 12 and 22, okay? So, if I increase . . . We just want to predict what will happen. So, if we increase the mouse's speed, okay? Without using the simulation, I want you to predict, by increasing the mouse's speed, will the cat still catch the mouse?

**Student B:** Well, if the cat can catch the mouse now, if we make the speed higher, I'm guessing the cat can't catch the mouse, because right now the speed is lower than the cat.

**Interviewer:** Okay.

**Student B:** If both . . . It was showing and, what, in comparison? . . . I was thinking, if we increase the speed, the mouse not only has a head start, but he has the advantage of more speed . . .That's why I think that the mouse would win.

Figure 8 captures how different phases described in the modeling cycle transpire in different phases of the model generated from the data. Note that the red circles in the diagram indicate running, understanding, and simplifying/structuring activities. These modeling activities fall under phase I, where participants worked with the simulation environment and explored its features and the results of various configurations. The blue circles in the diagram show "mathematizing" and "experience reconciliation" actions which occurred throughout exploration of the task as they attempted to structure their inquiry. The two green circles indicate activities of participants aimed at "interpreting" and "validating" their results.

## 6. Discussion

Our analysis of the participants' activities highlights two important components associated with using computer simulations: (1) the importance of a graphical user interface in helping students visualize the real-world problem and connect the simulation environment to the real-world problem, and (2) the differential modeling activities of students across the three stages of learning described in Section 4.3.

The graphical user interface of a simulation environment played an important role in helping students connect a computer simulation to the real-world problem it mimics. This can be in contrast to the simulations used in engineering, where simulations do not output a realistic animation of the real-world problem, but rather output graphs and statistics that report an output variable or measure the accuracy of the computer model in predicting ground-truth values of the simulation output. This further emphasizes the importance of realistic visualizations for elementary students. Furthermore, in all interviews students immediately connected the simulation environment to the real-world problem. None of the students mentioned that they struggled with the visualizations of the simulations, or that they did not see how the simulation connects to the real-world problem. This is reflected in our adjusted modeling cycle where the simulation replaced the situation model in the original extended modeling cycle [33].

The simulation replacing the situation model in the original extended modeling cycle [33] emphasizes another important distinction between learning mathematical modeling through interactive computer simulations and using computer simulations for developing a mathematical model. In all interviews, students construed the simulation as a tool that mimics the real world, without indicating the simulation is in fact using an accurate mathematical model itself to mimic the real-world problem. Indeed, there are two mathematical models involved: one used by the computer program to mimic the real-world problem, and one that students developed to explain the simulation outputs. In this study, all participants focused on discovering how simulation variables, which are real-world variables, affect the

real-world results visualized through the simulation. Previous studies have emphasized the importance of dynamic visualizations of computer simulations [31]. Our study shows that graphical visualizations are also essential for shielding students from the complex underpinnings of a computer program that realistically mimics a real-world problem.

We observed that the modeling activities students visit differ across the three stages of learning. For example, "understanding" was mostly observed in the exploration and discovery and structured inquiry stages of learning, while "reconciliation" and "interpreting" were most visited in the structured inquiry and synthesizing stages of learning. Previous studies have emphasized that students do not visit modeling cycle activities in order, and may take different paths, i.e., different sequences of modeling activities, to develop a mathematical model of a real-world problem. Our results show that in addition to differences across students, there are differences within students depending on their learning stage.

"Running" was the only activity that was frequently used in all stages of learning, but we observed differences in the utility of this basic tool across learning stages. "Running" was mostly used to learn to work with the simulation environment and understand the real-world problem during the discovery and exploration, was used in a structured approach to learn what the simulation output is under specific settings in the structured inquiry stage, and was used for confirming the predicted simulation output in the synthesizing stage. This differential use was also reflected in the simulation parameter selection, i.e., simulation design, as well. In the discovery and exploration stage, two students ran simulations without much consideration of the utility of the simulation setup in enriching their mathematical understanding, and only designed the next simulation setting they wanted to run. In the structured inquiry stage, students used a predetermined collection of simulation settings to collect data about the simulation, and in the synthesizing stage used one simulation setting for confirmation of their prediction.

An important connection between learning mathematical modeling through computer simulations and using computer simulations for developing a mathematical model is the exploration of real-world problems/simulations through trial-and-error-based experiments. When developing a mathematical model for a real-world problem, with or without technology, the modeler may start with understanding the real-world problem by conducting several experiments. Similarly, in our study, we observed that students ran several simulations as their initial experiments to learn about the simulation mimicking the real-world problem. This is consistent with the frequent use of the "running" activity during the discovery and exploration stage of learning.

Finally, we would like to emphasize that although we observed that all students benefited from exposure to the simulation, there were differences between students and across simulations. In this study, after unpacking students' modeling activities and learning, we focused on the common and consistent activities to learn about the role of simulations on students' modeling activities. However, the differences across students and the impact of different simulation activities on student learning are two important research questions that require further investigation in future studies.

## 7. Conclusions

The purpose of this study was to document and analyze modeling activities of three fifth-grade students in the presence of interactive computer simulations. The results indicate that graphical computer simulations helped the participants visualize the problem and learn about the effect of different variables. They allowed students to explore real-world problems and investigate the joint effect of multiple variables under various settings. Previously, Borromeo Ferri [38] offered that integrating technology allows learners to engage in modeling activities. She argued that different forms of technology can be useful in every step of the modeling process, particularly for interpreting and validating purposes. Our findings support these previously reported findings and offer that computer simulations served as an organizing tool for advancing learners' modeling activities.

Our results also offer a three-stage model of learners' interactions with the simulations. During the first stage, called exploration and discovery, students explored the environment to familiarize themselves with the computer environment, learned how they could interact with the simulation environment, and gained an initial understanding about the problem. In the second stage, called structured inquiry, students used the environment to observe the input–output relations under multiple settings and described their observations. They systematically used the simulation environment as a means to collect data and make an inference. They used their collective results to further investigate, discuss, and discover relations between variables. Finally, in the third stage, called synthesizing, students established laws that governed the behavior of the system and used them to make predictions about the behavior of the simulation setting prior to running it. They also explained why they were making certain predictions based on previous settings they had tested or rules they had inferred. Students moved between stages 2 and 3 to consolidate discrepancies between their predictions and simulation outputs until they were confident in their ability to explain how the system would behave without running the simulation.

As computers have become commonplace, students are now comfortable working with digital tools and can easily interact with such environments. Although mathematical modeling has been considered as an area of curricular emphasis in secondary school mathematics for years, elementary and middle school students are typically not provided enough modeling opportunities to develop mathematical understanding sufficient for applying their knowledge to real-world contexts. In particular, the use of computer simulations in early school years is unexplored and research on how elementary school students benefit from simulations in their early school years is in its infancy. In order to integrate simulations in the curriculum, several points may need to be considered. The simulation environment should be based on a real-world event, so that elementary students can easily relate to it. Also, simulation results should be presented in an appealing fashion, so that students can easily translate them to mathematical results, explaining the outcome of the system. Furthermore, it seems prudent to give students individual play time with the simulation environment in order to explore its properties to familiarize themselves with the simulation environment and learn how to manipulate its settings.

This study used semi-structured task-based interviews to explore students' modeling activities and interactions with simulation environments. However, this method had challenges. Some participants struggled to express themselves clearly. To address this issue, targeted questions were used to uncover their thinking, reasoning, and solution rationales across interview sessions.

While the current study shows that computer simulations are powerful tools that help students learn about complex real-world problems, we are cautious about the type of generalizations that might be drawn since our sample was limited to three participants. Further research is certainly needed in exploring the viability of our proposed model based on examining additional participants and across different grade levels. We also acknowledge that the number of simulations used in the current work was limited to three. Although not uncommon in research plans that focus on examining cognition and epistemology of mathematics, it is important to consider the growth of learners' modeling practices in the presence of simulation environments over a longer period of time and ways in which different simulations might influence dimensions of student learning according to their prior mathematical knowledge, exposure, and experience with technology or personal experiences with the contexts used in the simulations. Certainly, the role of teachers in guiding student learning in a classroom setting requires further investigation.

**Supplementary Materials:** The following supporting information can be downloaded at: https://www.mdpi.com/article/10.3390/educsci14040397/s1, Supplementary Table S1: Table of developed indicators and epistemic activities from the mathematical modeling cycle. Supplementary Table S2: Example of an event map (Cannon Ball simulation).

**Author Contributions:** Conceptualization, A.S. and A.M.; Methodology, A.S. and A.M.; Validation, A.S.; Formal analysis, A.S.; Investigation, A.S.; Data curation, A.S.; Writing—original draft, A.S.; Supervision, A.M. All authors have read and agreed to the published version of the manuscript.

**Funding:** This research received no external funding.

**Institutional Review Board Statement:** This study was approved by Behavioral and Social Sciences Institutional Review Board of The Ohio State University (study number 2018B0339 and date of approval 6 September 2018).

**Informed Consent Statement:** Informed consent was obtained from all subjects involved in the study.

**Data Availability Statement:** The data collected for this study is not publicly available due to privacy concerns of the participating students.

**Conflicts of Interest:** The authors declare no conflict of interest.

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
