# Peer review of "Interactive Learning: Unpacking the Influence of Computer Simulations on Students’ Mathematical Modeling Processes"

_education, doi:10.3390/educsci14040397_

Round 1

Reviewer 1 Report

Comments and Suggestions for Authors

See the attachment

Author Response

We thank the reviewer for their comments. We have revised our manuscript based on these comments, which has improved our manuscript. Briefly, as the reviewer suggested, we have extensively revised the “data analysis process” and “discussion” sections. As the reviewer suggested we merged the original discussion into the results section. The reviewer has brought up several important discussion points in their review, which we used to write a revised discussion section for the manuscript

Reviewer 2 Report

Comments and Suggestions for Authors

(lines 61-62) Research question is too ambitious. With only 3 students and 3 tasks is difficult to answer the posed question. Please, reformulate it.

Please, improve figures 1 and 2. If you print the paper, it's difficult to read the words inside the figures.

Figures 3,4 and 5 must be bigger. Again it's difficult to see in printed paper what is the content of the figures. 

Figure 7 of the MAD diagram is difficult to understand. The use of stars and time in seconds makes the overall reading difficult. Perhaps a diagram like the one found at https://mplac.blogs.uv.es/mad-ttt/ presented in

Pla-Castells, M., & García-Fernández, I. (2020). TaskTimeTracker: A tool for temporal analysis of the problem solving process. Investigación en Entornos Tecnológicos en Educación Matemática, (1). https://ojs.uv.es/index.php/ietem/article/view/16280

may be useful.

The interventions of the researchers are not clear during the presentation of the methodology; could they have influenced the student's resolution of the task?

Perhaps it would be interesting to introduce some category where the errors made by the students when performing the task are considered. 

Author Response

We thank the reviewer for the comments, which helped us to improve the manuscript. We have extensively revised the manuscript and have improved it based on reviewers’ feedback.

Round 2

Reviewer 1 Report

Comments and Suggestions for Authors

You can find my comments in the file attached

Author Response

Please find my response in the attached file. 
